# Tailoring the Spin Reorientation Transition of Co Films by Pd Monolayer Capping

**DOI:** 10.3390/nano14201662

**Published:** 2024-10-16

**Authors:** Benito Santos Burgos, Raúl López-Martín, José A. De Toro, Chris Binns, Andreas K. Schmid, Juan de la Figuera

**Affiliations:** 1Instituto Regional de Investigación Científica Aplicada (IRICA) and Departamento de Física Aplicada, Universidad de Castilla-La Mancha, 13071 Ciudad Real, Spain; raul.lopez@uclm.es (R.L.-M.); joseangel.toro@uclm.es (J.A.D.T.); christopher.binns@uclm.es (C.B.); 2Lawrence Berkeley National Laboratory, Berkeley, CA 94720, USA; akschmid@lbl.gov; 3Instituto de Química Física Blas Cabrera, CSIC, 28006 Madrid, Spain

**Keywords:** LEEM, SPLEEM, thin films, Cobalt/Palladium interface

## Abstract

We have characterized the magnetization easy-axis of ultra-thin Co films (2–5 atomic layers, AL) grown on Ru(0001) when they are capped with a monolayer of Pd. The addition of a Pd monolayer turns the magnetization of 3 and 4 AL-thick Co films from an in-plane to an out-of-plane alignment, but not that of a 5 AL-thick film. These observations are explained in terms of an enhancement of the surface anisotropy. The exposure of the sample to hydrogen, CO or a combination of both gases does not overcome this effect.

## 1. Introduction

Due to its ability to absorb hydrogen, Pd is an important material in gas sensor technology. Hydrogen absorption in bulk Pd leads to different phases depending on pressure and temperature [1,2,3,4]. When the H_2_ molecule is chemisorbed on the Pd surface, it dissociates into two H atoms, which diffuse into the Pd. In the initial stages, the so-called α-phase is formed (a disordered solid solution); when the amount of absorbed hydrogen increases, the β-phase is obtained, where the hydrogen atoms fill the octahedral positions inside the Pd crystal, producing a volume expansion of about 11% that may sometimes crack the crystal [2].

In Pd films grown on appropriate substrates, compressive stress due to substrate constraint can influence or suppress the formation of the β-phase.

For example, it was reported that hydrogen absorption appears to depend on the film thickness. In the limit of very thin films, the hydrogen solubility is limited or blocked [2,5]. Since the in-plane Pd lattice parameter is constrained to the substrate crystal, it limits the expansion of the Pd in this direction upon hydrogen absorption. Thus, hydrogen absorption can only influence lattice spacings along the film normal.

When Pd is grown on top of magnetic (ultra)thin films, such as Ni, Co or Fe, their magnetic properties can change significantly. Experimental results have shown that the surface-interface anisotropy and the strain are responsible for the perpendicular magnetic anisotropy (PMA) observed in these films [6]. Other experiments carried on Pd/Co have concluded that changes in both saturation moment and anisotropy are due to charge transfer to the Pd band at the Pd/Co interface when the films are exposed to hydrogen [1,7].

Moreover, structural details such as the amount of alloying, interfacial disorder and the presence of defects or diluted impurities at the interface have been identified as relevant factors in the study of the magnetism of palladium films exposed to hydrogen [8]. Some of these parameters can be controlled by the growth conditions, allowing additional control of the magnetic properties of the films.

The changes reported in the magnetic properties of the Pd/Co films upon hydrogen exposure are usually reversible, allowing the use of Pd/Co thin films as H_2_ sensors [1,2,9]. The technological interest of Pd/Co thin films is not limited to gas detection but also includes the magnetic recording industry [6,10]. These films show very large PMA and a large Kerr rotation; in addition, the magnetization easy axis can be controlled by the number of layers [11]. Initial hydrogenation produces an enhancement of the PMA [12]; however, further absorption yields a downward shift in the ferromagnetic resonance, which is consistent with a decrease in the PMA [13]. Interestingly, Schefer et al. showed that this shift in the PMA can be modified as a function of the number of Pd layers [14]. These results demonstrated a sensitivity of Pd/Co-based sensors down to 0.05% [8].

Finally, most of the above experiments were carried out exposing the films to molecular hydrogen or mixtures of different gases with molecular hydrogen. Here, we present a limiting case where only 1 AL Pd film is grown on Co films with different thicknesses. Next, the samples are exposed to both molecular and atomic hydrogen, which represents a valuable complementary study where H atoms (already dissociated) more easily diffuse to sub-surface sites in the Pd films.

## 2. Materials and Methods

The experiments were carried out in a spin-polarized low-energy electron microscope (SPLEEM) system, with a base pressure in the 10^−11^ Torr range. The instrument uses a spin-polarized electron source and a spin-manipulator that allows us to image the magnetization vector in the Co films. More details on this SPLEEM, spin-polarization control and vector magnetometry applications of the instrument can be found elsewhere [15,16,17]. The Ru(0001) single-crystal substrate was cleaned by flashing to 1700 K in a background pressure of 3 × 10^−8^ Torr of high-purity oxygen. Samples were flashed several times in the absence of oxygen before the Co deposition. Both Pd and Co were grown from electron-bombardment heated deposition sources. The typical flux rate was one atomic layer of Co in 3 min. A home-made atomic hydrogen source based on the cracker described by Bischler et al. was designed and built [18]. The cracker consists of a W tube 50 mm in length, 0.6 mm inside diameter, and 1.6 mm outside diameter supplied by Goodfellow with a purity of 99.95%. One side of the W tube was connected to the molecular hydrogen bottle through a leak valve and electrically isolated from the rest of the chamber by a macor^®^ tube. The other side of the W tube was heated up to 2000 K by electron bombardment. Molecular hydrogen was introduced into the W tube, dissociating at the hot end. For the final assembly, the W tube was mounted inside a water-cooled jacket. The efficiency of the cracker in the preliminary tests using a quadrupole mass spectrometer was 40% at pressures in the 10^−8^ Torr range. The hydrogen exposure in the experiments was determined from the pressure measured by the ion gauge and applying a correction factor of 0.42 as reported in [19].

## 3. Results and Discussion

### 3.1. Co Growth on Ru(0001) in LEEM

Cobalt films were deposited on Ru(0001) at 550 K. This favors the growth of extended regions with homogeneous thickness with a layer-by-layer growth mode despite the 8% misfit between Co and Ru bulk in-plane lattice parameters. A more detailed study of the growth, magnetism and structure of these Co films can be found in previous work [20,21]. At 550 K, Co grows by nucleating triangular islands on top of the Ru terraces, and all the islands have the same orientation on each substrate terrace; in adjacent terraces, the islands change their orientation by 180°. The islands grow and coalesce to form an atomic monolayer, and the nucleation of the second layer does not start until the first layer is completed.

The second layer also nucleates as triangular islands rotated by 180° with respect to the layer underneath. The third cobalt layer again grows as triangular islands (Figure 1d, but in this case the islands present two orientations on the same substrate terrace, which present very different magnetic properties [22]. The presence of two different orientations of the Co triangular islands on the same terrace is due to stacking faults in the Co islands, as reported previously [20,21,22,23]. The growth of additional layers up to 5 AL presents a similar behavior. The fourth layer starts by nucleating triangular islands with different orientations on the same terrace. Since the third layer already presents stacking faults, this gives rise to four different stacking sequences in the 4 AL Co islands. With the deposition of an additional layer, the 5 AL islands do not have well-defined triangular shapes, although the island density is not too different from the previous layer.

In agreement with our prior works [21,24], we find that only 2 AL-thick areas are magnetized in the out-of-plane direction, whereas other thicknesses (i.e., 1, 3, 4 and 5 AL) present an in-plane magnetization [21,22]. Figure 2a,b show LEEM and SPLEEM (with the magnetic sensitivity in the out-of-plane direction) images, respectively, of 2–3 AL films of Co. In Figure 2b, 2 AL-thick areas are the only ones that present black or white domains, corresponding to domains where the magnetization locally points out of the film (black) or into the film (white). The 3 AL-thick areas of the film do not show magnetic contrast out-of-plane. The last SPLEEM image (Figure 2c) shows the local magnetization along the in-plane direction. Now, the 2 AL areas have a zero magnetic contrast component in-plane, while the 3 AL areas display non-zero magnetic contrast.

### 3.2. Capping Co with Pd: Morphology

Figure 3a shows an image of a sample where 2–3 AL of Co were grown on Ru(0001) before the Pd deposition. At this electron energy (7 eV), the 3 AL-thick Co areas show a dark-gray contrast, while the 2 AL-thick areas appear medium gray. The sample shows several substrate steps where the 3 AL Co-thick areas formed a continuous ribbon, with additional 3 AL triangular islands with two orientations indicating the presence of hcp and fcc stacking sequences in the Co film, as explained above.

Pd was deposited while the substrate was kept at 550 K. The Pd islands appear as light-gray triangles on both the 2 AL and 3 AL areas of the Co film; see Figure 3b–d. The Pd seems to grow as triangular islands on top of the 3 AL Co islands with the same orientation as the underlying island. Only one orientation of the triangular islands is detected on each Co layer, suggesting that Pd does not present stacking faults with respect to the Co and follows an fcc stacking sequence. After completing 1 AL of Pd (Figure 3f), the contrast of the Co layers is reversed when compared with the bare Co film: Pd on 2 AL Co areas appears darker than Pd on 3 AL Co areas. Further Pd growth produced a uniform change in the electron reflectivity, but no further nucleation of islands was detected in real space by LEEM. Likewise, when Pd is grown on the thicker Co films (4–5 AL), the nucleation of individual Pd islands is not observed in LEEM, likely due to the growth proceeding by islands of sizes smaller than the resolution of the microscope, and only a uniform change in the electron reflectivity is observed during the Pd deposition.

### 3.3. Capping Co with Pd: Magnetic Effects

In addition to the LEEM real space images, SPLEEM images were acquired simultaneously during Pd deposition on the 2 + 3 AL Co film. Figure 4a displays the SPLEEM image with an in-plane spin-polarized electron beam corresponding to the same area shown in Figure 3a. The image shows magnetic contrast only in the 3 AL areas of the Co film, as explained before. As Pd is deposited (Figure 4c–f), the magnetic in-plane contrast of the 3 AL Co islands progressively disappears until no in-plane magnetic contrast is observed at the completion of a single Pd capping layer (Figure 4f). After the Pd deposition, the sample was cooled down to RT, and similar SPLEEM images were acquired with no contrast. Conversely, when the SPLEEM polarization was set out-of-plane (Figure 5b), both the 2 AL and the 3 AL Co areas presented out-of-plane magnetization, confirming that the deposition of a single monolayer of Pd switches the magnetization easy-axis of the 3 AL thick Co islands from in-plane to out-of-plane, while no change is observed in the easy-axis of the 2 AL areas.

Pd capping was also carried out on the thicker 4 + 5 AL Co films. Before the Pd deposition, both the 4 and 5 AL regions show in-plane magnetization, as expected for Co films thicker than 4 AL. Figure 6b,c show the SPLEEM images acquired with out-of-plane and in-plane polarization, respectively, after Pd deposition. While the Pd capping layer also switched the magnetization easy-axis of 4 AL Co from in-plane to out-of-plane, the magnetization easy-axis of 5 AL Co remained in-plane.

Figure 6d presents a graphical summary of our thickness-dependent SPLEEM observations before and after capping with palladium. Cobalt films exhibit out-of-plane anisotropy only up to 2 AL, falling in-plane for thicker layers (left panel). Capping with a Pd monolayer switches the magnetization easy axis from in-plane to out-of-plane (thus overcoming shape anisotropy) in the Co regions 3 and 4 AL thick. In other words, Pd capping “delays” the thickness-dependent transition from out-of-plane to in-plane anisotropy from 3 to 5 AL of cobalt.

Capping a magnetic film with a non-magnetic material produces two major effects [25]: (i) an increase in the coordination number at the interface yielding a reduction in the magnetic moment of the topmost magnetic layer, and (ii) electronic hybridization due to the wave function overlap of both materials. The latter decreases the magnetic moment not only of the topmost layer but of several layers under [26]. This mechanism plays a major role by altering the surface anisotropy and finally the magnetization of the sample. Although there are other effects involved, such as those produced by lattice distortions, they are typically reported to be much smaller than hybridization [11].

Thin films will typically show in-plane magnetization due to their strong shape anisotropy. However, the reduced symmetry of the system at the surface/interfaces often causes perpendicular anisotropy in ultrathin films. As anticipated before, our observations can be explained in terms of the competition between these mechanisms. The effective anisotropy is usually written as the sum of volume and surface contributions [25,27,28] (which depends inversely on the thickness) as
(1)$$K^{eff}=K^{v}+\frac{nK^{s}}{d}$$
where the factor *n* represents the number of identical interfaces and depends on the system under investigation [29,30]. The factor n is typically taken as 2 in multilayers and 6 in nanoparticles [27,31]. The volume contribution includes two terms: the magnetocrystalline $\;(K^{mc})$ and the shape anisotropy $\left(K^{shape}\right)$. Thus, Equation (1) can be rewritten as
(2)$$\;K^{eff}=K^{mc}-K^{shape}+\frac{nK^{s}}{d}$$

Cobalt presents high uniaxial magnetocrystalline constants [30] with *K*_1_ ≈ 0.4$3\times 10^{6}\frac{J}{m^{3}}$ [30,32,33], but the shape anisotropy is typically larger $\left(K^{shape}\;\approx \;1.3\times 10^{6}\frac{J}{m^{3}}\right)$ [30,32,33] and dominates the volume contribution to *K^eff^*. Using the parameters reported in the bibliography [28,30,32,33,34], we estimate a value of *K_v_* in the range of 700–790 $\frac{\mathrm{kJ}}{m^{3}}$.

The different signs of the terms in Equation (1) (reflecting different anisotropy directions, in- or out- of plane) yield the well-known spin reorientation transition (SRT) for *K^eff^* = 0, i.e., at a critical thickness of *t_cri_* =$\;\frac{K^{s}}{K^{v}}$ [25,27,35] Our experiments with uncapped Co show that the SRT takes place for the third Co layer. According to LEED-IV fits carried out by Gabaly et al. [20], the interlayer spacing for the three first Co layers on Ru(0001) is 2.05, 1.94, 1.99 Å, respectively. By taking an average value *K_v_* ~ 745 kJ/m^3^, we obtain that $K_{s}^{vacuum-Co}$ ~ 0.44 mJ/m^2^ in agreement with [34].

The value of the surface anisotropy for the capped films was similarly obtained, yielding $K_{s}^{Pd-Co}$ ~ 0.74 mJ/m^2^. Previous experimental reports yield results for $K_{s}^{Pd-Co}$ are in the range 0.16–0.92 mJ/m^2^ [6,36,37,38]. This variability in the result is attributed to the quality of the films: smooth epitaxial surfaces present high *K_s_* values (in the 0.58–0.74 mJ/m^2^ range) [11,36,39], while values as low as 0.16 mJ/m^2^ have been reported for films grown by sputtering Our results are also in agreement with theoretical calculations (0.66–1.1 mJ/m^2^) reported in the literature [28,38,40].

Figure 7 plots $K^{eff}d_{Co}$ as a function of the Co thickness (as derived from Equation (1)) obtained from the experimentally determined values of *K_s_* for the uncapped and Pd-capped cases. The negative slope indicates a negative volume anisotropy, i.e., favoring in-plane magnetization. The intercept at y = 0 gives the thickness at which the SRT takes place. Below this value, *K^eff^* is negative, indicating an in-plane magnetization; with decreasing thickness *K_s_* overcomes *K_v_* yielding out-of-plane magnetization (positive $K^{eff}$). Thus, this plot illustrates the effect of Pd capping, namely the enhancement of the value of *K_s_* by ~68%, resulting in a shift in the SRT of 1.95 layers, i.e., from the third layer to the fifth cobalt layer.

### 3.4. Effects of the Hydrogen Exposure

As we reported in a previous work, hydrogen absorption changes drastically the direction of the magnetization on the Co bilayer [21]. When the Co bilayer is exposed to 0.36 L ($1\;L=10^{-6}\;\mathrm{Torr}\cdot s$) of molecular hydrogen, the magnetization easy axis switches from an out-of-plane to in-plane direction. With this result in mind and considering the ability of Pd to both absorb hydrogen and increase the surface anisotropy, it is intriguing to see whether this SRT still takes place in Pd-capped Co bilayers exposed to hydrogen. To that end, we have tried to populate the Pd/Co interface with hydrogen by exposing the system to both molecular and atomic hydrogen. During the gas dosing, the magnetic domains of the Co film were monitored by SPLEEM in real time.

Since atomic hydrogen has one electron, its scattering power is much smaller than metal atoms, so a direct observation by typical electron spectroscopies is difficult. However, hydrogen absorption produces electronic changes in the metal atoms that might be simpler to detect than hydrogen itself. For example, it produces changes in the electronic structure of metals, such as a shift in the metal d band downwards in energy with respect to the Fermi level [41]. Also, the population of the subsurface with hydrogen increases the interlayer spacing of the last films, which can be detected by LEED-IV fits [42].

Low-energy reflectivity curves were acquired to monitor the adsorption of hydrogen. They have been proven to be a reliable method to detect the presence of different adsorbates on metallic surfaces, which leave characteristic fingerprints in the curves [41,42]. As a reference, curve (I) in Figure 8 corresponds to the bare surface of 1 AL Pd on 2 AL Co. The reflectivity is dominated by a broad peak around 22 eV. This peak is customarily attributed to the forbidden gap of the Bragg reflection [43]. The other oscillations detected in the range of 2–17 eV correspond to quantum size effects (QSE) due to interferences between electron reflection at the film surface and at the Pd/Co/Ru interfaces.

Figure 8 displays reflectivity curves after exposing the sample to different gases. After the hydrogen exposure [curve (II)], we observe that the reflectivity of the sample shows new features in the range of 7–18 eV, while the broad peak at 22 eV remains at the same position compared with the bare surface [curve (I)]. Similar changes have been reported in previous works where thin Pd films (2–6 AL) are exposed to molecular hydrogen, where the presence of hydrogen was reported mainly on the surface of the two first layers. Although most of the hydrogen was located on the surface, as was shown by LEED-IV fits [42], other experiments suggested that hydrogen can be dissolved in bulk Pd (111) even at room temperature and low hydrogen pressures (in the range of 10^−7^ Torr) [44].

Reflectivity curves also allow us to study changes in the work function of the surface upon hydrogenation, which shed light on the amount of the different absorbates. This work function change is obtained by comparing the energy (normally in the range of 0–5 eV), where the intensity drops after the gas exposure. After exposing the film to hydrogen, the work function of the surface increases by 0.21 eV. Similar increments were reported in different works dealing with hydrogen and Pd [45]. It has been established that the work function increases almost linearly with hydrogen exposure up to a coverage saturation value (θ ≈ 1) for a work function change of Δφ=0.20 eV [45]. Even at room temperature, it is possible to achieve hydrogen coverages in the range θ ≈ 0.8–1.0 for Pd (111) and Pd(100) [45]. Comparing our results to these reported Δφ values, we believe that hydrogen coverage in our sample must lie within that range. Although all the changes in the reflectivity curves, including the variation in the work function, strongly suggest the presence of the H atoms at the surface, even populating the interface, SPLEEM images acquired after and during the hydrogen exposure showed no changes in the magnetic easy axis of the Pd-capped Co bilayer capped, which remained out-of-plane.

Next, we replaced molecular hydrogen with atomic hydrogen, which provides a more efficient way to introduce hydrogen in the metal subsurface (since the molecule is already dissociated, the H atoms more easily diffuse to sub-surface sites in the palladium). Curve (IV) in Figure 8 is the reflectivity curve registered after exposing the sample to 90 L of H. It shows a small increment in the reflectivity, ca. 9–10 eV, unlike the case of molecular hydrogen, where the intensity increased clearly in the 7–10 eV range. In addition, the broad peak at 22 eV of the Pd/Co/Ru film vanishes and a new peak at 17 eV appears. All these changes in the electron reflectivity were also reported and studied in detail in previous works. The increase in the electron reflectivity in the range of 7–10 eV was identified, again, with the presence of hydrogen on the surface and the peak at 17 eV was ascribed to the presence of CO on the Pd surface [42]. The origin of this CO peak was previously reported in experiments carried out with the same type of hydrogen doser and is due to impurities that desorb from the tungsten tube at high temperatures [42]. This is also supported by the shift in the work function (1.07 eV), which compares very well with reported values of CO on Pd surfaces with a coverage of 0.5 AL. As in the case of hydrogen, the change in the work function is reported to be directly proportional to the CO coverages [46]. For CO on Pd surfaces such as (110), (211), (111), an increase in the work function between 0.7 and 1.25 eV is reported with a surface coverage in the range of 0.5–0.80 AL [46,47,48]. We thus estimate that the CO coverage in our sample must be close to θ ≈ 0.5.

Thus, our LEEM and LEED-IV experiments together with theoretical calculations we reported in previous works suggest that, when H and CO are both present at the Pd surface, hydrogen populates the surface first and then the subsurface positions, and finally, as the amount of CO increases on the surface, the CO removes the hydrogen from the surface by pushing it to the interface, which prevents the hydrogen desorption from the subsurface positions [42]. To test this mechanism, curve (III) shows the reflectivity curve acquired after dosing 3 L of CO on a Pd surface pre-saturated with molecular hydrogen. We observe that the peak at 9–10 eV (due to hydrogen adsorption) vanishes. At the same time, the broad peak at 22 eV shrinks and a new weak peak at 17 eV is detected, i.e., adsorbed CO displaces hydrogen from the Pd surface. Note how this curve shows an intermediate process between that of molecular hydrogen and atomic hydrogen exposure.

Comparing these results with those cited above, our experiments strongly suggest the presence of hydrogen in the interface between Co and Pd. However, after dosing 90 L of atomic H, SPLEEM images still showed no changes in the magnetization easy-axis. One possible explanation is that the amount of CO present on the surface is not enough, and it is not able to prevent hydrogen desorption from Pd. To check this hypothesis, a final experiment was performed. To force hydrogen into the Pd/Co sub-surface positions, we co-dose CO and atomic hydrogen, increasing in this way the amount of CO on the Pd surface. The partial pressures used were 3 × 10^−8^ Torr of atomic hydrogen and 2 × 10^−8^ Torr of CO, giving a total pressure of 5 × 10^−8^ Torr. The curve (V) shows the electron reflectivity of the surface after exposure to 60 L of CO + H. The curve is identical to that obtained when only atomic hydrogen was dosed: a small peak at 10 eV indicates that hydrogen is adsorbed on the surface, and a peak at 17 eV, together with the lack of a peak at 22 eV, signals the presence of CO on the Pd surface. Although all the data are consistent with hydrogen populating the sub-surface positions, no changes in the magnetization easy-axis were detected by SPLEEM after the combined CO and atomic hydrogen exposure.

In short, motivated by previous results where we demonstrated that it is possible to obtain a reversible SRT on 2 AL Co/Ru(0001) when the thin film is exposed to hydrogen, we have studied whether the same happens when the 2 AL Co thin is capped with Pd. Thus, we have tried to populate both the surface and Pd/Co interface with hydrogen to induce a change in the magnetization. After individual dosing of molecular hydrogen, CO or atomic hydrogen, no changes in the magnetization easy-axis were detected by SPLEEM despite the reflectivity data indicating the hydrogen incorporation at the Pd/Co interface.

As it was reported in previous works, the presence of any adsorbate on magnetic surfaces modifies the value of the surface magnetic anisotropy [49,50,51,52] yielding a corresponding shift in the critical thickness for the SRT [53].

Recent theoretical works demonstrated that small amounts of hydrogen in the Co/Pd interface enhance the PMA, but increasing the hydrogen concentration reduces the PMA, eventually flipping the magnetization [54]. On Fe films, the presence of hydrogen increases *K_s_* up to 30% [55]; however, when H or CO are adsorbed on nickel, they reduce the surface anisotropy between 30 and 50%, and the SRT critical thickness is shifted by 4 layers [56].

Some authors [30,50,53,57] divide the total surface anisotropy *K_s_* into two terms, $\frac{K^{s}}{t}=\frac{\left(K^{int}+K^{sur}\right)}{t}$ where *K^sur^* stands for surface anisotropy and *K^int^* is the interface anisotropy. When the sample was exposed to different gases, they showed that either the surface or the interface anisotropy can be affected by the presence of adsorbates and/or absorbates. Experiments carried out on Ni surfaces exposed to oxygen indicate that *K^sur^* is the more relevant term [50]. Similar conclusions were reported on surfaces exposed to H or CO, where the change in *K^sur^* was an order of magnitude larger than that in *K^int^*. These results suggest that the presence of hydrogen on the surface would be enough to drive a change in the *K_s_*, making the hydrogen absorption into the interface a secondary mechanism. Since our data strongly suggest the presence of hydrogen on the surface in all cases (displaced to the interface in the case of exposing the films to atomic hydrogen), we should expect a reduction in the value of *K^sur^* and/or *K^int^* resulting in a shift in the SRT to the second layer. Yet, the lack of such SRT in the SPLEEM images upon hydrogenation implies that the enhancement in *K_s_* due to the Pd overlayer overcomes the expected effect of adsorbates.

To confirm it, let us assume that hydrogen is present in the interface and/or on the surface and the value of *K_s_* is reduced by 50%, like in the case of Ni [53], i.e., down to *K_s_* = 0.34 mJ/m^3^. Using this value and that of *K_v_* employed in the previous section, *t_cri_* would shift to 2.5–3.0 AL, i.e., above the second Co layer; therefore, explaining the lack of in-plane magnetization in the second layer in our experiments. To observe a SRT in the second layer, *K_s_* should decrease by at least ~61% (from 0.74 mJ/m^2^ to 0.28–0.30 mJ/m^2^). But such reduction is higher than the values reported in the literature for hydrogen exposure.

Our findings guide future research seeking the switching of the easy-axis upon hydrogenation, which requires a selection of Pd and Co thicknesses leading to a smaller perpendicular anisotropy, namely stacks made of 1 AL of Pd on 3 AL Co/Ru (as indicated by the value of *t_cri_* extracted above) or 1 AL of Pd on 4 AL Co/Ru, for which *K^eff^* ~ 0 (ongoing experiments). In the latter case, the shape anisotropy and *K_v_* are predominant, and a small adsorbate-driven *K_s_* reduction would drive the SRT.

Finally, we should consider the difficulties of dosing a large amount of hydrogen under UHV conditions. Different works have reported that hydrogen can be absorbed by Pd in systems composed of ferromagnetic films capped with Pd layers. For example, in Pd/Co/Pd it was reported that upon hydrogen exposure, hydrogen was detected in the Pd films and the magnetization easy-axis of the Co underneath changed [12]. Other works [7] reported that hydrogen could be absorbed by Pd capping layers and it could diffuse through 3–4 AL Pd layers, inducing changes in the interlayer spacing of the films and producing an increase in the in-plane magnetic signal of the ferromagnetic film. The main difference between these experiments and ours is the hydrogen pressure used. The cited experiments were performed at hydrogen pressures around 1 atm (760 Torr), ten orders of magnitude higher than those employed in our UHV conditions. Although hydrogen absorption by Pd has been reported at high hydrogen pressures, the data shown in previous works related to Pd and hydrogen [42,58] indicate that under UHV conditions it is not possible to obtain significant hydrogen absorption through Pd layers even when using atomic hydrogen. Due to the limitations in the SPLEEM electron source, the maximum continuous dose was 90 L. Higher hydrogen pressures might be able to switch the Co magnetization easy axis.

## 4. Conclusions

We have studied the growth of 1 AL of Pd on 2–5 AL of Co/Ru(0001) by a comprehensive LEEM and SPLEEM characterization. A monolayer Pd capping delays the spin reorientation transition (SRT) for the Co system from 2 (uncapped system) to 5 (capped system) atomic layers as a result of the enhancement of the surface anisotropy. The system made by 1 AL Pd on 2 AL Co/Ru was exposed to both molecular and atomic hydrogen and monitored using low energy reflectivity curves. Even though the results suggested the presence of H at the Pd/Co interface, SPLEEM revealed no changes in the magnetization easy axis. This unexpected result, along with anisotropy energy arguments, led to the conclusion that the reduction in surface anisotropy due to gas adsorption is not enough to overcome the enhancement produced by the Pd capping monolayer.

## Figures and Tables

**Figure 1 nanomaterials-14-01662-f001:**
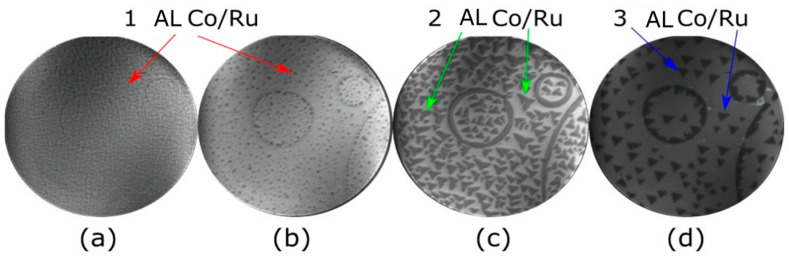
LEEM images acquired during the growth of Co on Ru(0001) at 550 K. (**a**,**b**) growth of the first layer. (**c**,**d**) completion of the second layer and nucleation of the third layer. The field of view (FOV) is 8 μm and the start voltage is 7.2 eV.

**Figure 2 nanomaterials-14-01662-f002:**
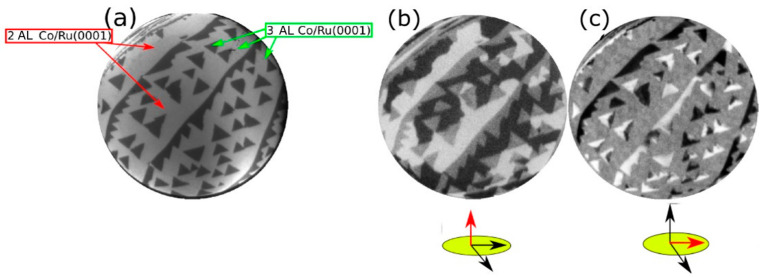
(**a**) LEEM image acquired at RT after growing 2–3 AL Co/Ru. (**b**) SPLEEM image with the electron-beam spin-polarization direction out-of-the plane. Magnetic contrast is observed only from the 2 AL thick areas. (**c**) Magnetic signal with the beam polarization along in-plane directions. The FOV is 8 μm and the start voltage is 7.2 eV.

**Figure 3 nanomaterials-14-01662-f003:**
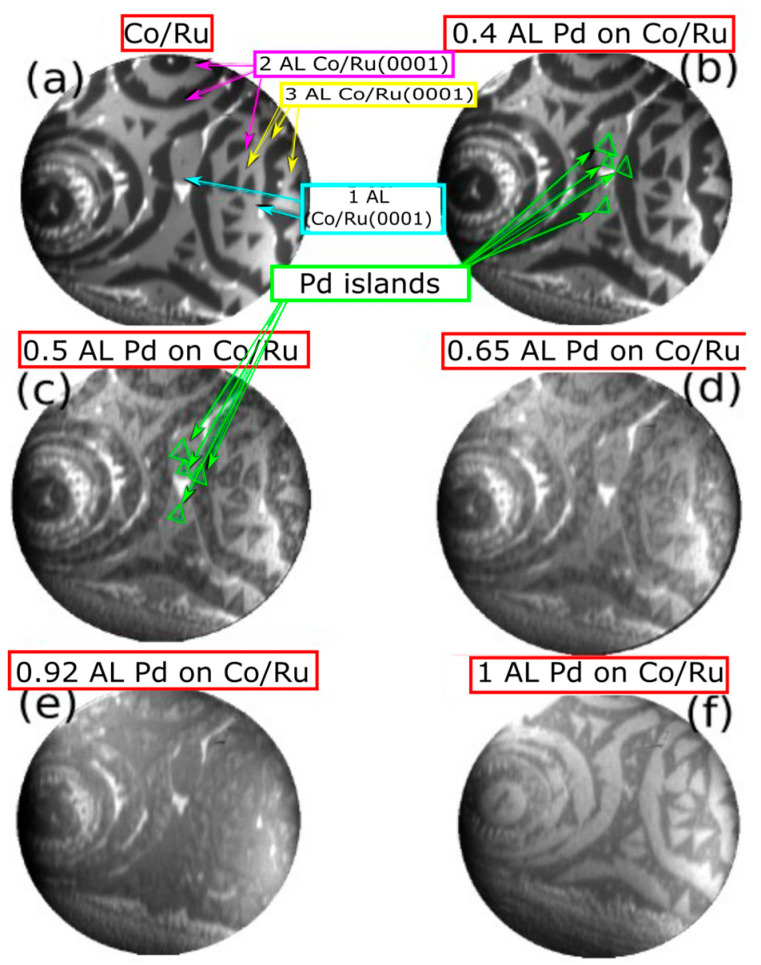
(**a**–**f**) Image sequence acquired in LEEM during the growth of 1 AL of Pd on a Co film with 1, 2 and 3 AL-thick areas exposed. The FOV is 8 μm and the start voltage is 7 eV. The growing Pd appears as light gray areas on the Co film.

**Figure 4 nanomaterials-14-01662-f004:**
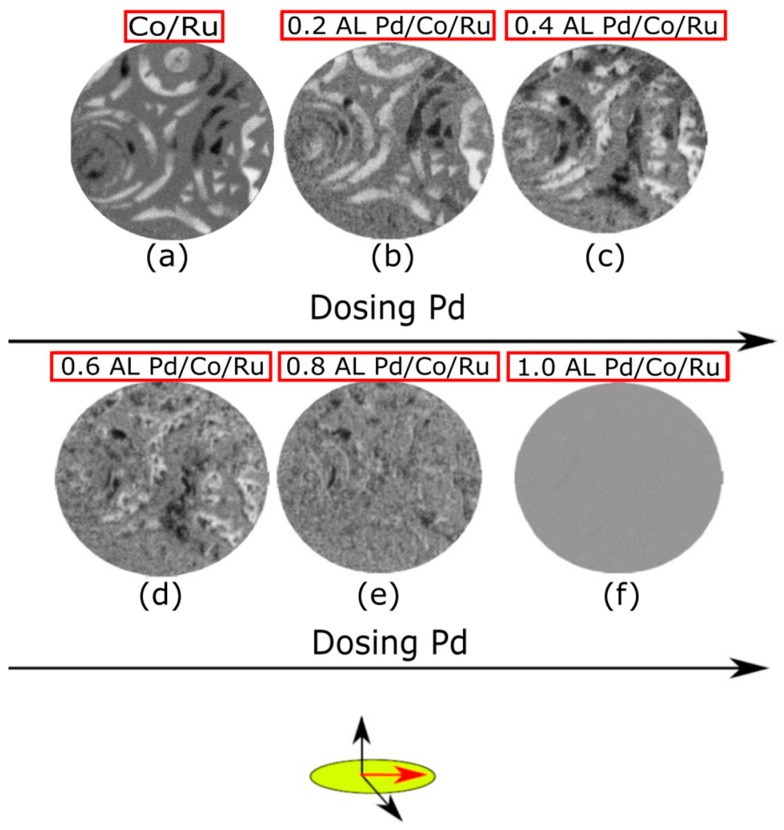
(**a**) LEEM image acquired on 2 and 3 AL-thick Co areas. (**b**–**f**) SPLEEM image sequence acquired upon Pd deposition showing the suppression of the in-plane magnetic contrast. The FOV is 8 μm. The start voltage is 7 eV for both LEEM and SPLEEM images.

**Figure 5 nanomaterials-14-01662-f005:**
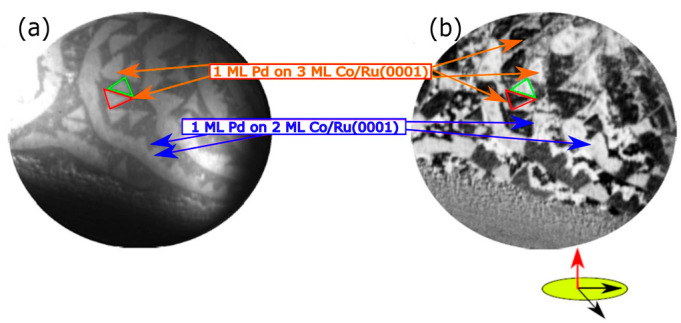
(**a**,**b**) LEEM and SPLEEM images, respectively, acquired at RT on the Pd-capped Co film presented in Figure 4. Both 2 and 3 AL regions present out-of-plane magnetization. The electron beam spin-polarization is out-of-plane. The FOV is 8 μm and the start voltage was 7 eV.

**Figure 6 nanomaterials-14-01662-f006:**
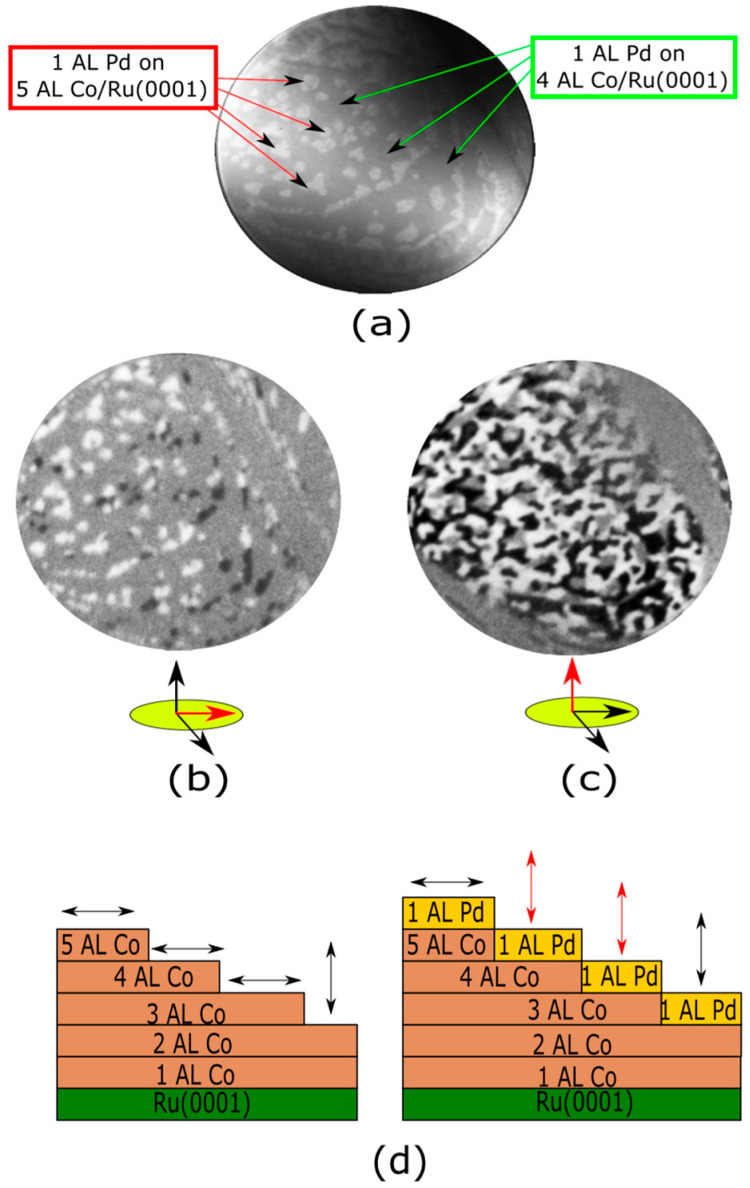
(**a**) LEEM recorded after growing 1 AL of Pd on areas covered with 4 and 5 AL of Co/Ru(0001). (**b**) SPLEEM image acquired with the electron beam spin-polarized out-of-plane. (**c**) SPLEEM image acquired with the electron beam spin-polarized in-plane. The FOV is 8 μm. The start voltage is 5.2 eV. (**d**) Schematic representation showing the magnetization easy axis before and after depositing the Pd capping. 2 AL, 3 AL and 4 AL Co areas display out-of-plane magnetization when covered with Pd, while in 5 AL areas the easy axis remains in-plane.

**Figure 7 nanomaterials-14-01662-f007:**
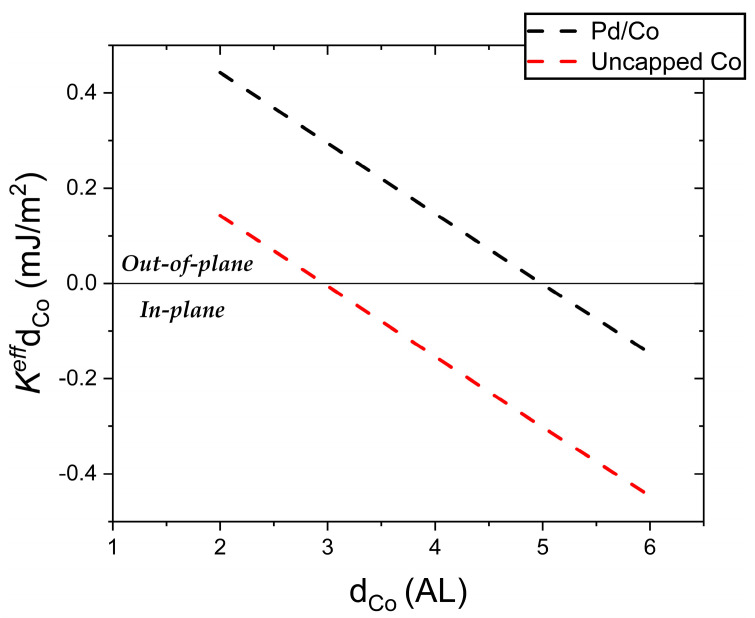
Effective anisotropy for the 6 first layers of Co as a function of the Co thickness, with and without a monolayer Pd capping. The data follows from the experimental determination of *K_s_*. The zero crossing indicates the critical thickness for the spin reorientation transition (SRT) in each case.

**Figure 8 nanomaterials-14-01662-f008:**
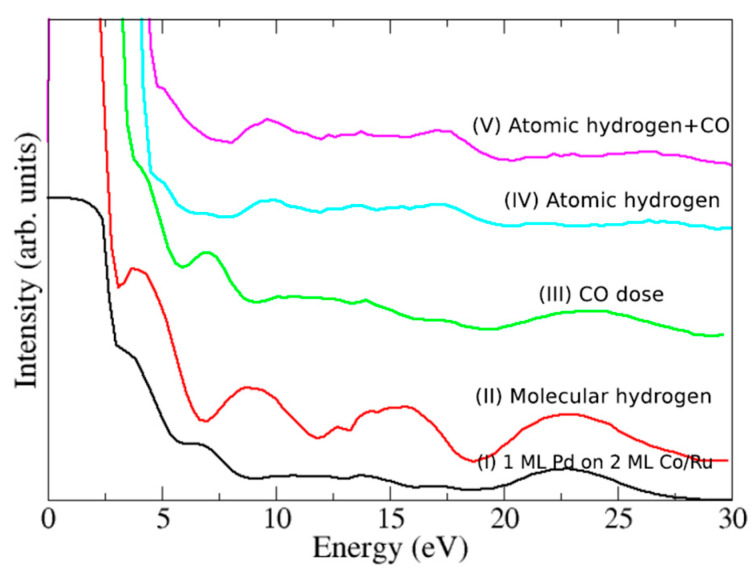
Low-energy electron reflectivity curves acquired on 1 AL Pd/2 AL Co/Ru after dosing different gases [(I) initial Pd/Co/Ru film], (II) 2 L_H2_, (III) 3 L_CO_ on a pre-saturated with hydrogen Pd surface, (IV) 90 L_H_, and (V) 60 L_H+CO_. The data have been offset for clarity.

## Data Availability

The raw data supporting the conclusions of this article will be made available by the authors on request.

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
