# Peer review of "Tailoring the Spin Reorientation Transition of Co Films by Pd Monolayer Capping"

_nanomaterials, 2024, doi:10.3390/nano14201662_

Round 1

Reviewer 1 Report

Comments and Suggestions for Authors

The study shows that adding a single layer of Pd to Co films on Ru(0001) significantly strengthens the surface anisotropy, delaying the spin reorientation transition (SRT) from 2 to 5 atomic layers. This means that Pd helps keep the magnetization aligned out-of-plane for thicker Co films. When the system (1 AL Pd on 2 AL Co) was exposed to hydrogen, there was no change in the magnetization direction, even though hydrogen appeared at the Pd/Co interface. This was unexpected because gas adsorption usually reduces surface anisotropy, which can affect magnetization. However, the Pd layer creates such a strong surface anisotropy that hydrogen alone cannot alter it.

This finding is important because it shows that Pd capping provides a stable magnetic alignment that can resist changes, even in environments with gas exposure, which is useful for developing durable spintronic devices. However, before considering the manuscript for publication, the authors should address the following questions:

1) Why did the authors choose the single-crystalline Ru(0001) substrate? Was it intended to deposit an epitaxial Co layer? If so, what is the out-of-plane orientation of the Co layer? Additionally, is the Pd layer epitaxial? What is its out-of-plane orientation? Is it Pd(111)?

2) Is an epitaxial Co/Pd layer, particularly Pd(111), essential for achieving perpendicular magnetic anisotropy?

3) Can the authors provide a discussion on the case of amorphous Co/Pd? Are the saturated magnetization and effective anisotropy constant (Keff) significantly smaller than those of the epitaxial Co/Pd bilayer?

Author Response

Comments1: Why did the authors choose the single-crystalline Ru(0001) substrate? Was it intended to deposit an epitaxial Co layer? If so, what is the out-of-plane orientation of the Co layer? Additionally, is the Pd layer epitaxial? What is its out-of-plane orientation? Is it Pd(111)?

Ru (0001) was chosen because it presents many advantages when growing thin Co films on it.  First, Co grows on Ru (0001) in a layer-by-layer mode up to 10ML, making it easy to identify the thickness of each layer. Another benefit is that we can obtain perfectly flat surfaces (3 μm in wide) where we can study the dynamics of the magnetic domains in real time. In addition, it has been reported that the amount of hydrogen absorbed by the Ru is negligible, which is a very useful property when dealing with hydrogen diffusion since it works as a barrier [2].

In previous work studying LEED-IV fits, some of the co-authors [1] showed that only the first Co layer is pseudmorphic with the substrate, while second and thicker layers relax very fast to the Co in-plane lattice parameter (2.50Å), being Co (0001) out-of-plane orientation.

The LEEM microscope used in these experiments does not allow us to acquire LEED diffraction patterns, or other structural characterization, but due to the big differences in the lattice parameter between bulk Pd (3.89 Å) and Co (2.50 Å) our guess is that Pd is not pseudomorphic with the Co.

 Since Co (0001) presents a hexagonal lattice, Pd (111) is the most suitable orientation in this case, but the mismatch between Co and Pd is still 10%  (Pd (111) lattice parameter is 2.75 Å)  suggesting that Pd is not pseudomorphic with the Co.

[1] El Gabaly F., Puerta J. M., Kelin Christof, et al. New journal of physics 9 (2007)

[2] O. Soroka a , J.M. Sturm a, et al. International Journal of Hydrogen Energy Vol 45,(2020)

Comments2: Is an epitaxial Co/Pd layer, particularly Pd (111), essential for achieving perpendicular magnetic anisotropy?

We believe that the orientation of the capping layer is not a key factor since the spin reorientation transition is due to electronic effects.

Comments3: Can the authors provide a discussion on the case of amorphous Co/Pd? Are the saturated magnetization and effective anisotropy constant (Keff) significantly smaller than those of the epitaxial Co/Pd bilayer?

The magnetocrystalline anisotropy (MCA) must be affected by the crystallinity of the films, since its value changes when the film is orientated with the crystal structure. On the other hand, in amorphous films, there is no preferred crystal direction, and we would find a random direction in the magnetization and probably a lower value in the MCA.

Since in our case, the SRT is due to electronics effects, we believe that the crystalline structure of the capping layer or the quality of the interface is not the most important factor, even with the presence of small amounts of alloy between Pd and Co we would obtain changes in the magnetization.

In the literature different mechanisms have been proposed as an origin of the PMA in amorphous films, such as atomic randomness, residual stress [4], the growth mechanism and even the sputtering pressure was pointed as a source of PMA in amorphous films [3]. Othe works, dealing with amorphous Tb-Fe films suggested that the total anisotropy increases when the number of Tb atoms per one Fe atom increases.

All these works suggest that the quality of the interface drives the anisotropy, but by means of MOKE measurements Beauvillain et al [5] reported the changes in the magnetization easy axis and the coercivity field in Co upon capping with Pd are obtained as soon as a capping layer starts to be deposited on the Co. Even for coverages in the sub monolayer regime this effect reaches values close to its maximum. Then it remains constant independently of the number of Pd layer deposited. 

We believe that electronic effect are much more important in our case.

[3] Lordan et al. Scientific Reports volume 11 (2021) 

[4] Ansar Masood et al. J. Phys. D: Appl. Phys. 57 (2024)

[5] P. Beauvillain, A. Bounouh, C. Chappert, et al. Journal of Applied Physics 76 (1994)

Reviewer 2 Report

Comments and Suggestions for Authors

This paper describes the spin reorientation transition on Pd/Co/Ru(0001) and investigates further SRT induced by surface modification using molecular and atomic Hydrogen. The same authors have published similar results on Co/Ru(0001), as cited in [22]. In this study, they extended their analysis to Pd capping and subsequent hydrogen adsorption/absorption. Although the Pd capping effect on SRT, stabilizing perpendicular magnetic anisotropy, is well known, the detailed investigation of the Pd effect on SRT using SPLEEM is still valuable. I recommend that this paper be published in Nanomaterials after some minor revision.

(1) Magnetic anisotropy changes by Hydrogen and Hydtrogn coverage in the subsurface

The author claims that the atomic Hydrogen at the Pd/Co interface does not change the interface magnetic anisotropy so much as in Ni. Is it possible to estimate the amount of subsurface Hydrogen in this study ? How much atomic hydrogen is necessary to induce SRT?

Author Response

Comments1: Magnetic anisotropy changes by Hydrogen and Hydrogen coverage in the subsurface

The author claims that the atomic Hydrogen at the Pd/Co interface does not change the interface magnetic anisotropy so much as in Ni. Is it possible to estimate the amount of subsurface Hydrogen in this study? How much atomic hydrogen is necessary to induce SRT?

In previous works [6], these authors, by means of LEEM, LEED-IV and theoretical calculations studied how the hydrogen is displaced from the surface of the Pd to sub-surface positions due to CO adsorption.  In the experiments, the surface was pre-saturated with 1MlL of hydrogen prior to the CO dosed.  Theoretical calculations and LEED-IV fits show an increment in the Pd interlayer spacing by 3% in agreement with β-phase, this would suggest a population between 0.5-0.7 of H in the sub-surface.

If β-phase is present in the systems, it would be not possible to increase the amount of hydrogen in the sub-surface at the pressure range used in these experiments as it was explained in the manuscript.

The change in the magnetization of the films after capping with Pd is mainly due to electronic hybridization effects [7], the presence other structural effects such as the amount of alloying, interfacial disorder, the presence of defects or diluted impurities in the interface have been identified as factors in other systems but the role of these structural effect are less than the hybridization between Pd and Co, in fact changes in the magnetic moments appears not only in the topmost films but of several layers under the surface.

We believe that increasing the amount of H in the sub-surface would not change the results, as we describe in the manuscript, the change in the anisotropy should be greater than those reported of oxygen or other gases.

[6] Cerdá J. I., Santos B, et al. The journal of physical chemistry letters J. Phys. Chem.Lett. 3 2012

[7] P. Beauvillain, A. Bounouh, C. Chappert, et al. Journal of Applied Physics 76 (1994)

Round 2

Reviewer 1 Report

Comments and Suggestions for Authors

The author has addressed all the questions I raised. The manuscript can be accepted in its current form.